# A Comparison Study of Deep Learning Methodologies for Music Emotion Recognition

**DOI:** 10.3390/s24072201

**Published:** 2024-03-29

**Authors:** Pedro Lima Louro, Hugo Redinho, Ricardo Malheiro, Rui Pedro Paiva, Renato Panda

**Affiliations:** 1CISUC, LASI, DEI, FCTUC, University of Coimbra, 3030-790 Coimbra, Portugal; redinho@student.dei.uc.pt (H.R.); rsmal@dei.uc.pt (R.M.); ruipedro@dei.uc.pt (R.P.P.); panda@dei.uc.pt (R.P.); 2School of Technology and Management, Polytechnic Institute of Leiria, 2411-901 Leiria, Portugal; 3Ci2—Smart Cities Research Center, Polytechnic Institute of Tomar, 2300-313 Tomar, Portugal

**Keywords:** music information retrieval, music emotion recognition, deep learning

## Abstract

Classical machine learning techniques have dominated Music Emotion Recognition. However, improvements have slowed down due to the complex and time-consuming task of handcrafting new emotionally relevant audio features. Deep learning methods have recently gained popularity in the field because of their ability to automatically learn relevant features from spectral representations of songs, eliminating such necessity. Nonetheless, there are limitations, such as the need for large amounts of quality labeled data, a common problem in MER research. To understand the effectiveness of these techniques, a comparison study using various classical machine learning and deep learning methods was conducted. The results showed that using an ensemble of a Dense Neural Network and a Convolutional Neural Network architecture resulted in a state-of-the-art 80.20% F1 score, an improvement of around 5% considering the best baseline results, concluding that future research should take advantage of both paradigms, that is, combining handcrafted features with feature learning.

## 1. Introduction

Most early attempts at Music Emotion Recognition (MER) tackled classical machine learning (ML) techniques, where much of the effort is put into feature engineering [1,2,3,4]. The usual pipeline for improving the classification of such techniques involves identifying gaps in musical dimensions, such as melody, harmony, rhythm, dynamics, tone color (timbre), expressivity, texture, and form, designing feature extraction algorithms that can capture those dimensions, and then training ML models on those extracted features. However, due to the complexity involved in the process, most current works only employ low- and mid-level descriptors, many proposed for other problems of the broader Music Information Retrieval (MIR) field. One recent exception is the work by Panda et al. [5], with the development of new emotionally relevant features based on audio analysis, which resulted in 76% accuracy in the 4 Quadrant Audio Emotion Dataset (4QAED) dataset. The study aimed to create new features to break the current MER glass ceiling as observed in the MIREX challenge, where results attained a plateau of about 69% accuracy [5]. However, the design process of such features is a time-consuming and challenging task that requires expert domain knowledge in signal processing, musicology, and ML.

Deep learning (DL) has recently seen a rise in popularity for its ability to reduce such workloads due to its ability to learn relevant features from raw input data automatically and has been applied in a variety of fields. Recently, various DL methods have been applied to tackle MER, many of which employ Convolutional Neural Networks (CNNs), Recurrent Neural Networks (RNNs), and various combinations of the two [6,7,8]. Typically, raw input data are represented by a spectrogram, but end-to-end architectures that do not require previous processing have also been proposed [9,10]. In addition, learning paradigms, such as transfer learning from other domains with larger available datasets [11,12], and different data representations, such as from embeddings that can be extracted from pre-trained CNNs [13] have also been proposed.

Despite the potential seen in the field of Computer Vision, these techniques have limitations, such as the need for large amounts of quality labeled data, a common problem since the infancy of the MER field. Classical ML methodologies have previously dealt with this problem by applying audio transformations to the available samples and obtaining new synthesized samples to increase the training set for the chosen algorithms. Since previous studies on this matter focused especially on singing voice [14] and genre recognition [15], the impact of data augmentations specifically for MER is not well known and needs to be assessed.

A drawback of methodologies based on neural networks is their lack of interpretability given their black-box nature, meaning that it is not known what kinds of features deemed relevant for the data are learned and extracted during the training process. For the case of MIR, questions have arisen in the past regarding whether these networks are learning relevant information for the task at hand, such as genre, with the same concerns applicable to emotion.

However, a study by Choi et al. [16] shows that a five-layer convolutional portion of a CNN learns to extract features closely related to melody, harmony, percussion, and texture for four very different songs through a process called auralization. More recently, Won et al. [17] demonstrated that a self-attention mechanism is able to learn relevant information for instrument, genre, and emotion detection using heatmaps to visualize which areas of the spectrograms are taken into account to perform classification.

Taking into account the various promising paths to exploit DL-based approaches, in this article, we conduct a comparison study of various classical ML and DL methodologies applied to MER to understand the effectiveness of these techniques, using the 4QAED dataset complemented with a recent expansion. Methodologies include architectural improvements, the inclusion of audio augmentation techniques, experimenting with alternative input data representations, and exploiting knowledge from related tasks. Moreover, the expansion of the baseline dataset enabled the study of the impact of dataset size on the classification accuracy of DL models.

The output of this study resulted in the following contributions: (i) an ensemble of a Dense Neural Network (DNN) and a CNN architecture, which resulted in a state-of-the-art 80.20% F1 score (based on data augmentation); (ii) a thorough comparison between possible methodological improvements for solving MER; and (iii) an analysis of the impact of dataset size and class balancing on classification performance.

## 2. Background

The connection between music and emotions has long been a focus of research in music psychology. Emotion from a musical piece can be examined through the lens of (i) expressed, or the emotion the composer or performer tries to convey to the listener; (ii) perceived, or which emotion is identified by the listener; and (iii) induced, or the emotion felt by the listener. These different types of analyses may produce equal or completely different interpretations of the emotional content of a song, but a key difference lies in the different levels of subjectivity [18]. Perceived emotion has been shown to provide the highest level of objectivity among the types as mentioned earlier and can be found as the focus of most works in the MER literature.

Various models have been proposed to represent the spectrum of human emotion, either by clustering similar emotions, also designated as categorical models, such as Hevner’s Adjective Circle [19], or by having a multi-dimensional plane where the axes represent different biological systems to mimic how the brain perceives emotion, intuitively referred to as dimensional models in the literature, the most widely accepted being Russell’s Circumplex Model [20], seen in Figure 1.

Many scholars have raised concerns about both categories of models. On one hand, categorical models do not realistically reflect the continuous nature of the emotional spectrum, leading to limitations in pinpointing the exact emotion. On the other hand, dimensional models are known to have a high degree of complexity because of the basis on which they are constructed, and although they may provide more accurate accounts of the emotions reported by annotators, prior knowledge of their inner workings is required to properly do so, severely impacting the range of annotators using such models and the accuracy of the output annotations [21].

Recently, Panda et al. [5] proposed the 4QAED dataset using labels from experts found on the AllMusic API [22]. Through a thorough process, these labels were translated into arousal and valence values, collectively called A–V values, the *y*- and *x*-axes of Russell’s model, respectively. Instead of maintaining the continuous approach of this model, all annotations were grouped into one of the four quadrants, making them discrete and more easily understood as categorical models. A more in-depth explanation of the dataset, as well as its expansion, is provided in the following section.

## 3. Methods

This section describes the methodologies explored in this work, ranging from architectural improvements to alternative data representation, data augmentation techniques, and knowledge transfer.

We begin by defining both ML and DL baseline methodologies, discussed in more detail in Section 3.1, and evaluating them on multiple datasets. The obtained results provide a comparison point with the explored methodologies, in addition to making it possible to assess the impact of increased dataset size and class imbalance.

The remaining section explains the explored methodologies and what led us to consider them. These include architectural improvements that exploit time-related information (Section 3.2.1), architectures that learn features from portions of whole samples (Section 3.2.2), alternative input representations obtained through high-dimensional projections (Section 3.2.3), increased training data through sample synthetization (Section 3.2.4), and exploiting learned information from related tasks (Section 3.2.5).

### 3.1. Baseline Architectures

As a baseline for our experiments, we first considered the state-of-the-art model from Panda et al., a simple Support Vector Machine (SVM) classifier (classical baseline) in which hyperparameters were fine-tuned for each dataset experimented using the same set of optimal features found in the original work.

A CNN architecture based on the work by Choi et al. [6] (see Figure 2) was previously developed by our team and is used as the DL baseline. The original architecture was adapted so that, instead of outputting a binary vector, the extracted features are processed on a small DNN that predicts one of the four quadrants from Russell’s model. This baseline is essential for assessing the viability of new DL architectures on our datasets and provides a basis for further improvement. The Stochastic Gradient Descent (SGD) optimizer was used to train the DL baseline, and the following hyperparameters were found to be optimal: *batch size* = 150, *epochs* = 200, *learning rate* = 0.01. An early stopping strategy was employed, which halted training when the accuracy of the train set reached a value above or equal to 90%, as it overfits above this value as found from previous experimentation. These points are the default configuration for the remaining approaches described in this section unless explicitly stated otherwise.

### 3.2. Explored Methodologies

We began by reviewing recently proposed DL approaches for MER. It is important to note that this work focuses on improving the classification of static emotion (Static MER) in music. We do not delve into emotion variation detection (MEVD), a higher complexity problem based on identifying the emotional content and its fluctuations across an entire music piece, or other modalities such as lyrics.

Recently, Won et al. [23] conducted a comparison study on various DL architectures, including the Convolutional RNN (CRNN) architecture, an end-to-end approach, a simple architecture that takes small segments of the whole sample as input, and an architecture with trainable harmonic filters. Implementations for all of the abovementioned are available in a GitHub repository [24], which we adapted for experimenting with our data. The remainder of this section briefly describes the explored approaches, including existing and novel ones.

#### 3.2.1. Architecture Improvements

As a starting point to improve our baseline architecture, two Gated Recurrent Units (GRUs), reported in their original paper to be more stable to train than Long Short-Term Memory units [25], were added to our baseline CNN architecture in an attempt to process and extract time-domain-specific features. To understand how appropriate the CNN portion of this network is for such a task, an implementation of the CRNN architecture, depicted in Figure 3 was adapted from the aforementioned repository.

In addition, one of the best-performing methodologies was a simple ensemble of the baseline CNN with a DNN fed with all the extracted features, previously pre-trained and with its weights frozen, that fuses the information before being post-processed by a smaller DNN. It was decided to fuse information from both networks at the feature level to understand how handcrafted and learned features complement each other. As stated before, the reason for the lack of improvement in classical approaches is missing features relevant for emotion recognition. With the inclusion of the learned features from the CNN portion, we should observe how relevant these are in relation to the handcrafted features.

To understand the impact of information fusion at the feature level, we first conducted experiments using only a DNN architecture. The full set of 1714 features was considered, as well as the top 100 features used for training the SVM baseline. The best-performing model is incorporated into the previously described ensemble. The experimented architectures are depicted in Figure 4.

The idea is to combine the information extracted from both approaches to improve the overall classification. To improve the capabilities of the CNN portion, we pre-train it with synthetic samples resulting from classical audio augmentation techniques (time shifting, time stretching, pitch, and power shifting, as discussed below) already studied in the same work, referred to as Hybrid Augmented for clear distinction. The architecture is depicted in Figure 5.

#### 3.2.2. Segment-Level Approaches

Our previous work focused on using the full 30-s samples available on 4QAED as the model’s input. However, humans can identify emotions in smaller samples with some ease. Considering the small size of the datasets used for evaluating the explored methodologies, breaking down these samples into smaller segments has the added advantage of increasing the number of training examples, an indirect form of data augmentation. By considering small inputs at a time, the network is also able to learn local-level features more easily when compared with sample-level approaches. A simple model that applies this idea is presented in [23], referred to as ShortChunk CNN. The architecture is presented in Figure 6. To train the model, each segment was treated as its own sample. In contrast, for testing, the mode of all segments’ predictions pertaining to a sample is used as the final prediction, also known as a many-to-one approach. The best hyperparameters values found were: *batch size* = 50, *epochs* = 100, *learning rate* = 0.001.

Another usual architectural component in previous DL works is using a set of convolutional layers to downsample and extract features from spectral representations, requiring the definition of parameters for generating such a representation. Although the ideal parameters have been previously studied as is the case in [6], they are not architecture-independent. A solution to this problem would be to work directly with the raw audio signal without pre-processing and extracting features directly from it. This was achieved by Lee et al. [9] who proposed a model referred to as Sample CNN, which uses a sequence of one-dimensional convolutional blocks, very similar to the two-dimensional variant, and processes the outcome in a dense layer. The architecture is depicted in Figure 7. With these architectures, the best values for the hyperparameters were almost the same as those for the ShortChunk CNN, with the exception of the number of epochs, which increased to 150. It is important to note that the original models were designed to output one of a set of labels, differing depending on the dataset used, and were translated from PyTorch to TensorFlow with reworked output to categorical labels.

#### 3.2.3. Data Representations

As mentioned previously, when describing the Sample CNN architecture, Mel-spectrograms may not be the optimal representation for training a model to classify emotions. Embeddings, or the mapped representation of a sample in a lower-dimensional space learned from the original data’s space, are very popular in Natural Language Processing (NLP) tasks, such as for Speech Emotion Recognition (SER), due to the natural translation of words to smaller dimensions. The same idea was applied to audio by Koh et al. [13], utilizing the OpenL3 deep audio embedding library (v0.4.1) [26] and training the classical ML techniques classifier on its output. The embeddings are obtained directly from a Mel-spectrogram representation, resulting in a feature matrix of 298 × 512.

Results were provided for the baseline dataset for this study, reaching a 72% F1 score using the Random Forest (RF) classifier of the scikit-learn library (v1.0.1) [27], very close to the classical baseline. The experiment was replicated and extended to the baseline dataset extension. The embeddings provided by the autoencoder mentioned when describing the DeepSMOTE-like approach in the following subsection were also tested for comparison.

#### 3.2.4. Data Augmentation

We further explored both classic and DL approaches for data augmentation. For the former, several audio augmentation techniques were applied directly to the audio signal of a sample, randomly increasing or decreasing a factor associated with the transformation, namely, time shifting (shifts start or end by a maximum of 5 s), pitch shifting (increasing or decreasing pitch by a maximum of 2 semitones), time stretching (speeding up or slowing down by a maximum of 50%), and power shifting (increasing or decreasing amplitude by a maximum of 10 dB). Continuing in this line, we experimented with more of these techniques using the audiomentations library (v0.24.0) [28], namely, the following:Time–Frequency Masking (TFM), popular in the field of SER, which applies a mask over a portion of the time and frequency domains [29];Seven-Band Parametric Equalization (SB), applying a seven-filter pass on the sample, changing its timbre in the process;Tanh Distortion (TD), applying a distortion similar to an electric guitar;Random Gain (RG), randomly increasing or decreasing the loudness of a sample;Background Noise (BG), which adds random background noise from a specified set of samples, in our case, the ESC-50 dataset [30].

For each transformation, a random value is picked from a set of predefined intervals to be used as the factor for the transformation, e.g., RG predefined interval is between [−12.0, 12.0] dB. These intervals were left unchanged from the defaults found in the library. It is important to note that a transformation is only applied to each sample once. This means that when experimenting with a single transformation, the training data are effectively doubled, while for the previously discussed Hybrid Augmented approach, the training data are increased fourfold since we are applying four transformations at a time.

As for DL-based techniques, Generative Adversarial Networks (GANs) [31] were previously tested with underwhelming results. Not only is the process of training a GAN overly complex when compared with classical audio augmentation but the lack of constraints when sampling the learned space from the data leads to noisy and emotionally ambiguous samples.

To impose some constraints on the generation of samples, the SMOTE [32], or Synthetic Minority Oversampling Technique, was considered. Although it was apparent that directly applying this technique to the raw audio signal produces even noisier samples than the GAN, owing to the high dimensionality of the audio signal, we used the autoencoder used for training the GAN to reduce significantly the number of dimensions of a sample akin to the DeepSMOTE approach proposed by Dablain et al. [33]. A raw sample in a waveform representation presents approximately 482 k values or dimensions to represent a 30 s sample with a 16 kHz sampling rate. In contrast, by passing the Mel-spectrogram representation through the autoencoder, we retrieve an embedded representation comprised of 60,416 values, a significant decrease for improving the SMOTE’ing process. To the best of our knowledge, this is the first application of the technique to music samples.

One problem with this approach is the choice of SMOTE implementation because many alternatives exist, many of which have domain-specific applications. Regarding which is the most optimal SMOTE variant to use, the article by Kovács [34] as well as the accompanying repository (v0.7.1) [35], are a comprehensive resource to better support a decision, presenting a comparison of over 80 variants. Because of this large number, we only experimented with the most widely used variants, SMOTE, BordelineSMOTE, and Adasyn. BorderlineSMOTE, specifically the Borderline_SMOTE2 implementation, was found to be the best fit based on preliminary tests. In addition, it was found from these tests that 25 synthesized samples for each quadrant, in addition to the original ones, were optimal, with such an increase accompanied by an increased batch size of 200.

As a final note, precautions are taken to prevent synthesized samples from leaking to the test set. It is possible that by modifying the samples, the same also happens to the underlying emotion. For example, when we apply pitch shifting with a +2 factor, i.e., an increase of 2 semitones, to a melancholic song, we may be making the song happier. Manual re-annotating the synthesized samples is not at all feasible due to the necessary resources, and such efforts should be directed to new original samples that can increase the dataset as a whole.

To ensure that the synthesized samples do not distort the evaluation of the model, we first assign each original sample to the train or test set, and only after the synthesized samples are added to the train set, only if the corresponding original sample is already present. This guarantees that no synthesized sample is used for evaluation and preserves the viability of the evaluation metrics. With this in mind, the benefits of using a data augmentation technique can be assessed indirectly by the performance of the model in question. If it increases, we can infer that the techniques involved are beneficial for our tasks and that they most likely preserve the original emotion, while if they considerably change the emotion, we would observe a decrease in performance.

#### 3.2.5. Transfer Learning

Another approach is to transfer the learned knowledge from a domain with a larger data corpus to deal with the reduced size of the dataset, which in practice means transferring the learned weights from a network to a new network with a different task, freezing them to avoid information loss, and replacing the output portion of the model appropriate for the task at hand. Our team previously experimented with exploiting the learned weights of a network trained for genre recognition for MER. Here, the idea was not to use a larger dataset but to take advantage of the learned information pertaining to genres to improve emotion recognition since specific genres are tightly connected to particular emotion quadrants, e.g., heavy metal and Q2, reggae, and Q4 [36].

In a similar fashion, we experimented with transferring the knowledge from the models presented by Park et al. [12] developed for artist classification. For the purposes of this work, the simpler model was adapted, consisting of a sequence of 5 one-dimensional convolutional blocks, a global average pooling layer, and a dense layer that outputs a 256-value vector, as seen in Figure 8. For the experiment, the model’s weights, which can be retrieved from the article’s accompanying repository [37], were loaded and frozen, and the last layer was replaced with an also dense layer outputting to one of the quadrants. Differences from the DL baseline configurations include using the Adam optimizer in place of SGD, as per the original implementation. Moreover, almost identical hyperparameters were used, except for a decrease in the batch size to 100.

Another experiment was performed to understand the impact of applying the information gained from larger datasets for MER, using the available weights for the CRNN model trained on the MagnaTagATune (MTAT) [38], MTG-Jamendo (JAM) [39] and MSD dataset on Won’s repository, referred to as CRNN TL. It is also important to note that these weights result from training the CRNN to output for the available set of labels, i.e., multi-label classification. The optimization process here is adaptive, meaning that it changes at certain epochs, beginning with Adam with a learning rate of 0.001 until epoch 80, then changes to the SGD optimizer with a learning rate of 0.0001, decreasing to 0.00001 at epoch 100, and finally to 0.000001 at epoch 120. The authors state that this leads to a more stable training process and ensures optimal results at 200 epochs with only a batch size of 16, both of which are used as these hyperparameter values, in addition to reducing the necessary computational resources for model optimization.

## 4. Evaluation Details

In this section, we introduce the datasets used for evaluating the presented methods (Section 4.1), data pre-processing details (Section 4.2), and the experimental setup used to conduct evaluation (Section 4.3).

### 4.1. Datasets

As mentioned in Section 2, the dataset used for the conducted experiments is the 4QAED dataset [40] previously created by our team [5]. The dataset contains 900 samples evenly distributed among the four quadrants of Russell’s model. Each corresponds to a set of emotions: Q1 represents happiness and excitement; Q2, anger and frustration; Q3, sadness and melancholy; and Q4, serenity and contentment. The dataset provides 30-second excerpts of the complete songs and two sets of emotionally relevant handcrafted features as data sources. The two sets of features contain (i) 1714 found to be relevant for emotion recognition, (ii) and the top 100 features obtained after feature selection. Regarding the targets, the dataset provides categorical labels for one of the four quadrants.

As part of this work, the dataset in question was expanded, increasing the number of available samples from 900 to 1629. Henceforth, each dataset is referred to as Original-4QAED and New-4QAED. Furthermore, as can be seen in Table 1, besides the complete (C), unbalanced, New-4QAED dataset, a balanced subset (B), comprising 1372 samples, was also experimented with. The latter also takes into account the distribution of genre in each quadrant to avoid possible bias.

### 4.2. Data Preprocessing

To obtain the Mel-spectrogram representations of the samples used as input data for these methodologies, the librosa (v0.8.1) [41] Python library was used with default parameters. One exception is the sample rate, which was set to 16 kHz after experimenting with different values.

Although higher sample rates are normally used due to more accurately presenting auditory information, the resulting Mel-spectrograms are significantly more computationally heavy for the model to process. Other studies have also found that DL-based architectures are robust to the decrease in information related to lower sample rates [42].

### 4.3. Experimental Setup

The performed experiments were conducted on a shared server with two Intel Xeon Silver 4214 CPU with a total of 48 cores running at a clock speed of 2.20 GHz as well as three NVIDIA Quadro P500 with 16 GB of dedicated memory, the latter necessary for developing and evaluating each network in a reasonable time. Due to high demand at the time of evaluation, Google Colaboratory [43] was also used, where it offered a very similar GPU and either a NVIDIA P100 PCIE with 16GB or NVIDIA T4 with the same amount of dedicated memory depending on availability.

Most of the experimented DL approaches were developed using the TensorFlow’s (v2.8.0) [44] Python library, allowing us to build and optimize complex models in a simple and quick manner. The PyTorch (v2.0.1) [45] library was also used to utilize the provided weights for the pre-trained CRNN models discussed in Section 3.2.5.

## 5. Experimental Results and Discussion

Results for each methodology and considered datasets are presented according to the high-level division discussed in Section 3.2.

The presented metrics are Precision, i.e., how many samples of a given class are predicted as this class, Recall, i.e., how many samples are correctly predicted as belonging to a given class, and F1 score, i.e., the harmonic mean between Precision and Recall. These are obtained through the widely used scikit-learn Python library [27].

The evaluation process to obtain these metrics consisted of firstly optimizing the relevant hyperparameters on Original-4QAED, experimenting with a set of possible values in a grid search strategy to serve as a baseline for performance on New-4QAED, and utilizing these same parameters to ensure a fair comparison.

For each set of hyperparameters, a 10-fold and 10-repetition stratified cross-validation strategy is used, totaling 100 different train–test splits, as it is the accepted approach to deal with the small dataset sizes and provide reliable results. For each repetition, the dataset is randomly split into 10 different portions while ensuring equal distribution of quadrants as found in the original dataset, using 9 portions for training and 1 for testing. The portion held out for testing changes for each train–test split, resulting in 10 different combinations for each repetition. An example of the process for obtaining the train–test splits can be seen in Figure 9.

The hyperparameters’ values tested using this method differed from methodology to methodology. For those based on the baseline CNN, neighboring values were tested to account for possible variations in the data. Otherwise, the same process was followed using values from the original articles if available, using the baseline CNN values as backup. Although a more thorough analysis would require that the same process be repeated when changing the dataset, this was not possible due to resource constraints. Regardless, conclusions can be drawn from the impact of different sizes and quadrant distributions of a dataset.

A decision was also made regarding the multiple classical audio augmentation techniques and multiple datasets on the evaluated CRNN TL methodology to proceed with the evaluation on New-4QAED only if the performance on Original-4QAED at least matched the DL baseline methodology. This is reflected in the absence of results in the tables presenting the results of data augmentations and transfer learning methodologies across datasets.

Regarding observed improvements, the increased dataset size was beneficial for the baseline CNN with GRU and CRNN methodologies, which saw an increase from 60.07% to 61.99% and 60.35% to 63.33% in F1 score, respectively from Original- to New-4QAED C, both better in relation with the DL baseline results (see Table 2), as seen in Table 3. It was also apparent that increased dataset size made the optimization phase more stable than previously observed. There was a slight decrease when the balanced variations of the latter were applied, reinforcing the importance of the dataset size.

As for the DNN-based methodologies, the 1714 feature set model performs better on the New-4QAED variations, while the 100 feature set performs considerably better on the Original-4QAED. This is to be expected since the top 100 features were found using the latter and may not translate to a dataset with more samples. Thus, using the complete feature set for our Hybrid Ensemble should perform better since the DNN is able to process the relevant features for a given dataset.

To wrap up the improvements related to architectures, the overall best result was obtained with the Hybrid Augmented methodology, which reached an F1 score of 80.20% on the balanced subset of New-4QAED. Here, both the size and quadrant distribution heavily influenced the obtained score, the latter most likely related to the biased nature of the DNN, similar to classical ML techniques.

Some improvements were also observed when applying Time–Frequency Masking, Seven-Band Parametric Equalization, and Random Gain, which achieved the best results with an increase of around 1.5% in F1 score, seen in Table 4, compared with the DL baseline on Original-4QAED and was consistently better on New-4QAED. As for the Tanh Distortion and Background Noise transformations, their poor results may be caused by considerable changes in the underlying emotion when compared to the original samples. These results call for a need to conduct more studies on data augmentation applied to MER, as most of the applied techniques in the literature are drawn from studies in other fields, as already discussed in Section 1, with implications for the emotional content of the resulting samples not being known.

In a more negative light, all segment-level methodologies performed poorly compared to the DL baseline as presented in Table 5. Such poor performance may be attributed to the reduced size of the datasets compared with the ones used in the original proposal of the architectures, which are already in the order of hundreds of thousands of samples, which means that the available training data thwart our own, and also the difference of the problem-solving approach as already mentioned in the previous section. Moreover, splitting samples into smaller segments may introduce more variability to the data and, in turn, make it difficult for the architecture to learn relevant features for discerning each quadrant, a hypothesis that should be further investigated.

Other methodologies, especially related to knowledge transfer and data representation, performed worse than this baseline as seen in Table 6 and Table 7, respectively. In regard to knowledge transfer, both approaches presented significant underperformance compared with the same baseline, which implies that this information is not useful for emotion recognition, particularly regarding the multi-label classification approach when using larger datasets. The poor performance of these methodologies may be attributed to significant differences from the learned features for the specific task, meaning that potentially relevant information is lost due to a higher prevalence of features not relevant for emotion recognition. Other possible factors include the quality of the datasets considered for pre-training the models, especially MSD, and the data distribution in terms of emotion, genre, and other relevant factors for MER. Experimenting with an ensemble of models trained for emotion recognition and another related task should be considered in the future.

As for embedding-based methodologies, we were not able to replicate the results presented for the OpenL3 embeddings on Original-4QAED, reaching, at most, a 55.70% F1 score against the reported value of 72%, which may be due to the unclear data splitting (apparently, the authors followed 80/10/10 train–validation–test data splitting instead of 10-fold cross-validation). Moreover, the parameters disclosed in the original approach lacked mention of the parameters for creating the RF classifier, so it was understood as using the default parameters from the scikit-learn implementation. At the same time, cross-validation was another point not made clear, for which we applied the usual method for consistency matters. We also observed that the autoencoder embeddings performed consistently better on New-4QAED when compared with OpenL3 embeddings, which may indicate that these are not the best suited for MER.

The poor results of the autoencoder embeddings were also reflected in the DeepSMOTE-based augmentation, with no significant improvement over the DL baseline. The lack of improvement may be attributed to the high dimensional embedding space, as sampling from this space provides little variability in comparison with the original samples. Another possibility is the distortion of important regions in the Mel-spectrogram representations, which make it difficult for the network to classify the synthesized sample. Reducing the input data size, e.g., using the segments of the full samples, should decrease the embedding space dimension and produce more relevant synthesized samples.

## 6. Conclusions and Future Directions

In this study, the performance results of different classical ML and DL methodologies were evaluated on differently sized datasets to assess the impact of data quantity for various approaches, with a greater focus on the latter to deal with the existing semantic gap found in the former approaches. Various routes have been explored, including improvements to previously developed architectures and exploring segment-level ones, applying data augmentation to increase the available training data, performing knowledge transfer for leveraging information from other datasets and/or domains, and using different data representations as input.

From the evaluated methodologies, the proposed Hybrid Augmented, an ensemble of both a CNN trained with synthesized samples in addition to the original ones, and DNN using Mel-spectrogram representations and previously extracted features from each song as input, achieved the best result overall of an 80.20% F1 score on the New-4QAED balanced dataset. Another significant improvement was obtained by applying the CRNN on the increased sized New-4QAED datasets, surpassing the DL baseline by approximately 2% on the complete set, and the improvements observed when applying classical data augmentation.

The comparison between the various methodologies has also highlighted the performance improvement provided by classical audio augmentation techniques in addition to the already discussed architectural improvements. The same cannot be said regarding segment-level architectures, knowledge transfer from related tasks, and embedding-based input representations, although some of these may be improved as already discussed in the previous section. It was also evident from the obtained results that dataset size is more impactful than class balance for classification performance in most cases, which we can observe in the CRNN experiment for instance, where the New-4QAED complete set outperforms the balanced set by around a 1.5% F1 score. Moreover, this is more noticeable in the segment-level architectures experiments, where the complete set outperforms the balanced set by 4%.

The results indicate that research should be pursued to develop novel classical features and improve DL architectures for further performance improvement. Moreover, data augmentation research specifically for MER appears to be a promising route to fully exploit DL models’ abilities to extract relevant features automatically. With increasing training data, future DL architectures should incorporate an RNN portion to extract time-domain-specific features. To conclude, various spectral representations as inputs are also an exciting research route as found from early experimental efforts, but it is necessary to address the unstable nature of such approaches first.

## Figures and Tables

**Figure 1 sensors-24-02201-f001:**
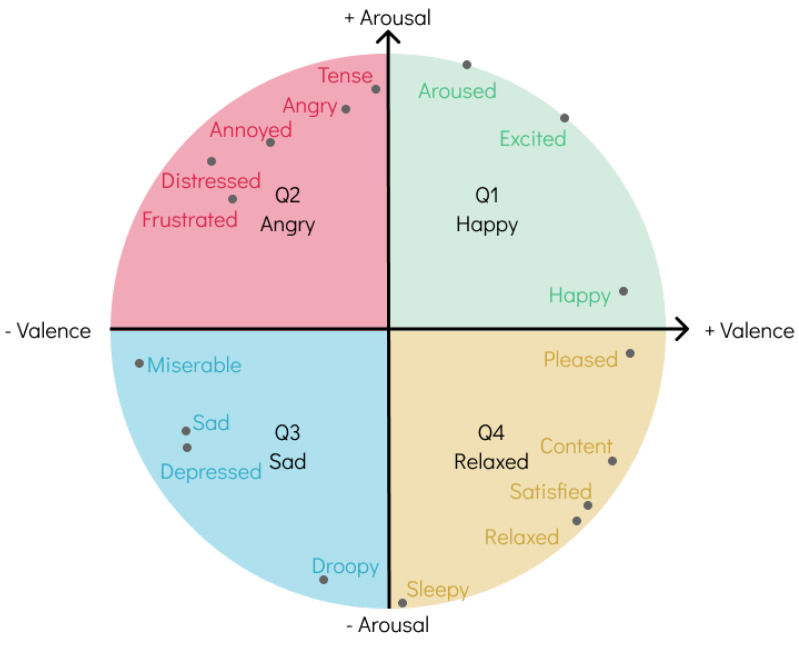
Russell’s Circumplex Model. Emotions can be mapped with continuous values as shown by the words in each isolated point, or as discrete labels, representing a broader emotion.

**Figure 2 sensors-24-02201-f002:**
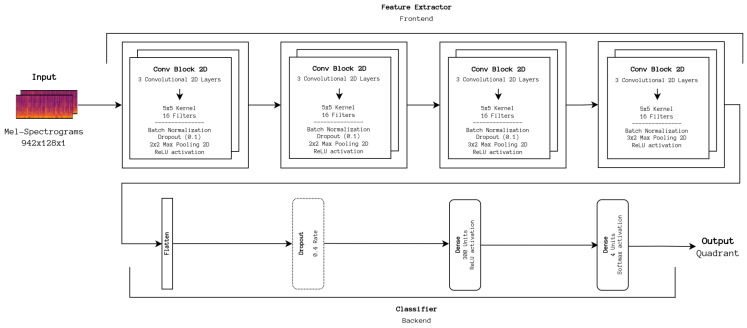
DL baseline architecture. The frontend portion first extracts relevant features inferred from the input data, which are then fed to the backend for classification.

**Figure 3 sensors-24-02201-f003:**
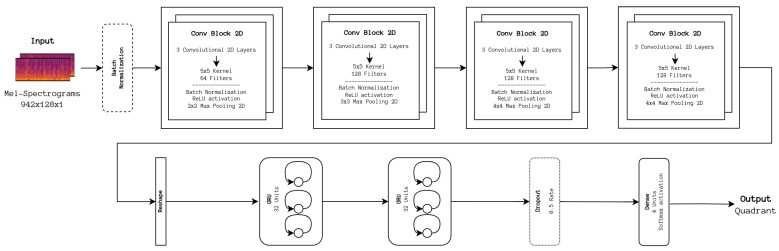
CRNN architecture. The number of filters applied to the input data is larger when compared with the DL baseline architecture, and, as a result, the extracted information is more heavily downsampled. In addition, the backend portion replaces the dense network with two GRU units to process time-related information.

**Figure 4 sensors-24-02201-f004:**
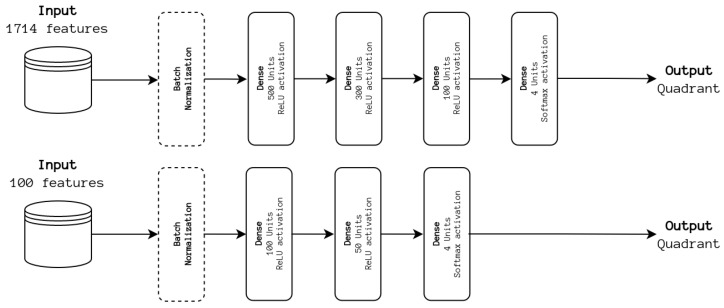
DNN architectures. The input feature sets are processed, akin to a feature selection process, and classified.

**Figure 5 sensors-24-02201-f005:**
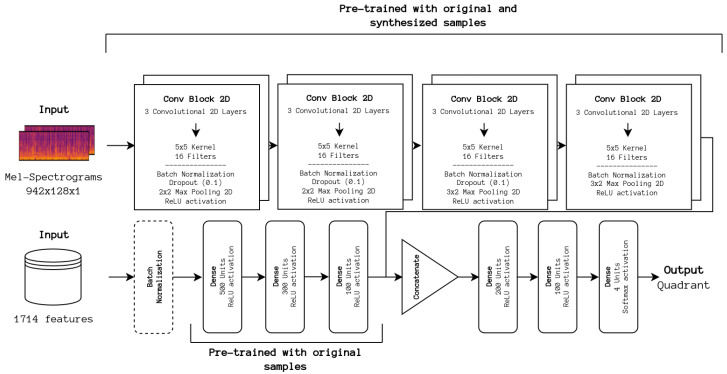
Hybrid Augmented architecture. Both feature extraction portions are pre-trained with the train set samples and synthesized samples for the DL feature extraction portion exclusively. Late feature fusion is performed before classification.

**Figure 6 sensors-24-02201-f006:**
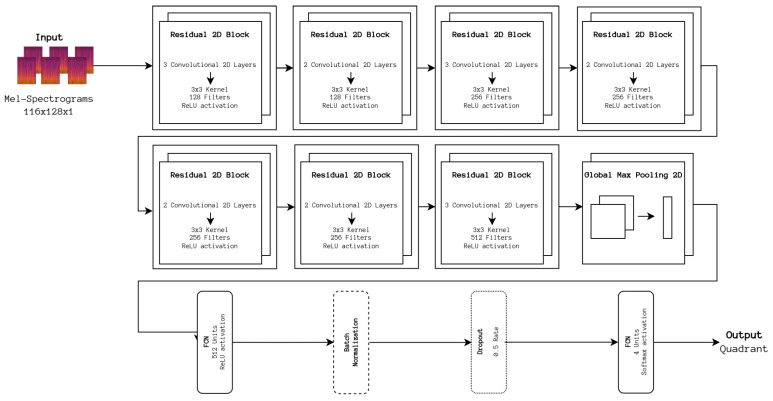
ShortChunk CNN architecture. The model processes smaller chunks of a full sample at a time, increasing the data available for training. The full sample is classified by aggregating the smaller chunks’ predictions.

**Figure 7 sensors-24-02201-f007:**
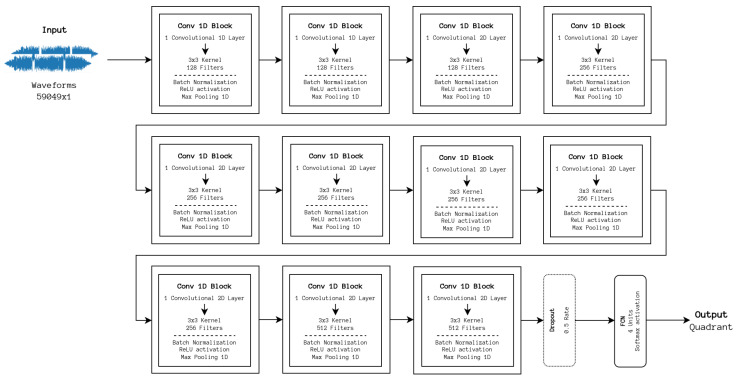
Sample CNN architecture. The process for classifying samples is similar to ShortChunk CNN; however, the features used for classification are learned directly from the raw audio sample.

**Figure 8 sensors-24-02201-f008:**
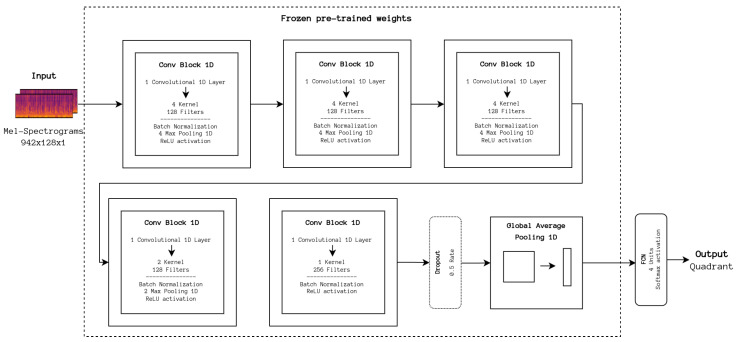
Architecture of CNN pre-trained on artists classification task. The feature extraction portion is frozen; only the classification portion is trained.

**Figure 9 sensors-24-02201-f009:**
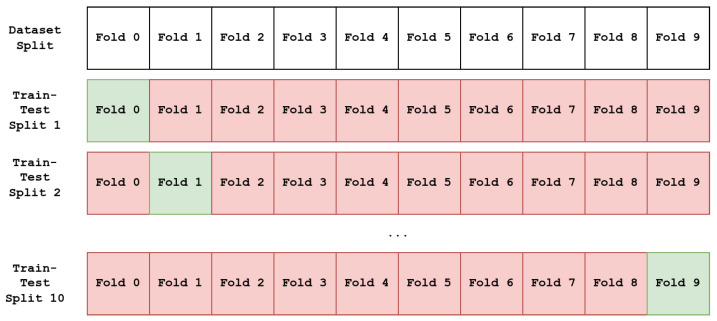
Example of 10-fold stratified cross-validation process for obtaining train–test splits. Red folds are part of the train set, while green folds are the test set. Each fold retains the original class distribution of the full dataset.

**Table 1 sensors-24-02201-t001:** Datasets used for evaluation with respective sample distribution.

Dataset	Q1	Q2	Q3	Q4	Total
Original-4QAED	225	225	225	225	900
New-4QAED C	434	440	397	358	1629
New-4QAED B	343	343	343	343	1372

**Table 2 sensors-24-02201-t002:** Precision, Recall and F1 score of baseline methodologies across datasets.

Methodology	Metrics	Original-4QAED	New-4QAED C	New-4QAED B
SVM Baseline	Precision	75.63%	69.92%	70.03%
	Recall	76.03%	70.26%	70.05%
	F1 Score	75.59%	69.79%	69.82%
DL Baseline	Precision	61.60%	62.46%	61.39%
	Recall	61.21%	63.99%	63.42%
	F1 Score	60.62%	61.66%	60.28%

**Table 3 sensors-24-02201-t003:** Precision, Recall, and F1 score of methodologies with improved architectures across datasets.

Methodology	Metrics	Original-4QAED	New-4QAED C	New-4QAED B
Baseline CNN With GRU	Precision	61.58%	62.29%	60.69%
	Recall	61.01%	62.46%	60.01%
	F1 Score	60.07%	**61.99%**	58.85%
CRNN	Precision	65.14%	64.20%	63.31%
	Recall	65.07%	64.03%	63.34%
	F1 Score	**64.63%**	**64.09%**	**62.54%**
DNN	Precision	69.41%	69.01%	68.58%
With 1714	Recall	69.27%	69.00%	68.40%
Feature Set	F1 Score	**69.18%**	**68.63%**	**68.05%**
DNN	Precision	72.61%	67.63%	67.77%
With 100	Recall	72.74%	67.67%	67.72%
Feature Set	F1 Score	**72.48%**	**67.40%**	**67.41%**
Hybrid Augmented	Precision	67.81%	68.15%	80.56%
	Recall	68.08%	68.14%	80.50%
	F1 Score	**68.04%**	**67.85%**	**80.24%**

Bold results indicate statistical significant improvements over the DL Baseline.

**Table 4 sensors-24-02201-t004:** Precision, Recall, and F1 score of methodologies trained with synthesized data across datasets.

Methodology	Metrics	Original-4QAED	New-4QAED C	New-4QAED B
Baseline CNN	Precision	63.05%	62.51%	62.33%
With Synthesized	Recall	62.75%	62.17%	61.85%
Samples (TFM) [29]	F1 Score	**62.03%**	**61.82%**	**61.39%**
Baseline CNN	Precision	63.38%	62.54%	62.13%
With Synthesized	Recall	62.79%	62.16%	61.71%
Samples (SB)	F1 Score	**62.12%**	**61.73%**	61.01%
Baseline CNN	Precision	63.37%	63.02%	62.35%
With Synthesized	Recall	63.13%	62.80%	62.10%
Samples (RG)	F1 Score	**62.24%**	**62.08%**	**61.36%**
Baseline CNN	Precision	61.83%	N.A. *	N.A. *
With Synthesized	Recall	61.58%	N.A. *	N.A. *
Samples (TD)	F1 Score	60.59%	N.A. *	N.A. *
Baseline CNN	Precision	61.97%	N.A. *	N.A. *
With Synthesized	Recall	61.79%	N.A. *	N.A. *
Samples (BG)	F1 Score	60.84%	N.A. *	N.A. *
Baseline CNN	Precision	61.91%	62.40%	61.62%
With Synthesized	Recall	61.61%	62.02%	61.41%
Samples (DeepSMOTE) [33]	F1 Score	60.70%	61.47%	60.48%

Bold results indicate statistical significant improvements over the DL Baseline. * Experiment was not conducted for this dataset.

**Table 5 sensors-24-02201-t005:** Precision, Recall, and F1 score of methodologies with segment-level architectures across datasets.

Methodology	Metrics	Original-4QAED	New-4QAED C	New-4QAED B
ShortChunk CNN [23]	Precision	64.66%	64.07%	60.23%
	Recall	61.48%	62.13%	59.19%
	F1 Score	60.61%	61.84%	57.07%
Sample CNN [9]	Precision	62.64%	65.17%	62.43%
	Recall	61.26%	62.62%	56.70%
	F1 Score	60.92%	60.78%	54.46%

**Table 6 sensors-24-02201-t006:** Precision, Recall, and F1 score of methodologies with embedded data representations across datasets.

Methodology	Metrics	Original-4QAED	New-4QAED C	New-4QAED B
OpenL3 Embeddings [13]	Precision	55.67%	53.92%	53.03%
	Recall	56.75%	54.49%	53.18%
	F1 Score	55.70%	53.62%	52.85%
Autoencoder Embeddings	Precision	50.63%	53.78%	53.56%
	Recall	50.40%	55.45%	54.76%
	F1 Score	50.18%	53.56%	53.69%

**Table 7 sensors-24-02201-t007:** Precision, Recall, and F1 score of methodologies leveraging knowledge transfer across datasets.

Methodology	Metrics	Original-4QAED	New-4QAED C	New-4QAED B
CNN	Precision	51.95%	51.81%	51.56%
Pre-Trained On Artist	Recall	53.93%	53.29%	52.43%
Classification Task	F1 Score	50.85%	50.27%	50.22%
CRNN	Precision	51.93%	52.97%	52.16%
Pre-Trained On	Recall	51.71%	53.72%	52.50%
MagnaTagATune	F1 Score	50.21%	51.70%	51.44%
CRNN	Precision	49.98%	N.A. *	N.A. *
Pre-Trained On	Recall	48.07%	N.A. *	N.A. *
MGT-Jamendo	F1 Score	47.94%	N.A. *	N.A. *
CRNN	Precision	47.50%	N.A. *	N.A. *
Pre-Trained On	Recall	46.18%	N.A. *	N.A. *
MSD Subset	F1 Score	45.84%	N.A. *	N.A. *

* Experiment was not conducted for this dataset.

## Data Availability

4QAED Dataset: http://mir.dei.uc.pt/downloads.html (accessed on 27 March 2024).

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
