# Peer review of "A Comparison Study of Deep Learning Methodologies for Music Emotion Recognition"

_sensors, 2024, doi:10.3390/s24072201_

Round 1

Reviewer 1 Report

Comments and Suggestions for Authors

The topic of the article is quite interesting, there are not many research studies on recognizing emotions in music, which is due to objective reasons, such as the lack of large datasets, the presence of copyrights on works. The article provides a good overview of the research, the article is well structured. At the same time there are a number of comments:

1. It is not clear from the text of the article how many examples were obtained for each class after augmentation, such information would allow estimating the size of the augmented dataset. The augmentations used were "pitch shifting (increasing or decreasing pitch by 2 semitones), time stretching (speeding up or slowing down by 50%)", such transformations can result in a change of the class of the samples. Have either experiments or expert evaluation of the augmented data been conducted to match emotion classes?

2. Experiments use k-fold validation, this approach is accepted in modern research. But from the text of the article we can understand that the whole dataset was augmented first, and then the partitioning into folds took place. With this approach, the original example may end up in the training set, and the transformed one in the test set or vice versa. In such a case, training and testing take place on almost the same data, which can artificially increase the quality of classification. 

3. In the paper, the authors compare their results with the results from the article "Panda, R., Malheiro, R., Paiva, R. P.: Novel Audio Features for Music Emotion Recognition. IEEE Transactions on Affective 469 Computing, 11(4), pp.614-626 (2020)", while not citing the result of research in the last 3 years. In the work "Sá, Pedro Marques Alegre de. MERGE Audio: Music Emotion Recognition next Generation-Audio Classification with Deep Learning. MS thesis. 2021." the f1 value reaches 88%, with the paper "Liao, Yi-Jr & Wang, Wei-Chun & Ruan, Shanq-Jang & Lee, Yu-Hao & Chen, Shih-Ching. (2022). A Music Playback Algorithm Based on Residual-Inception Blocks for Music Emotion Classification and Physiological Information. Sensors. 22. 777. 10.3390/s22030777." accuracy value of 92.07%.

Author Response

We sincerely thank the reviewer for the positive assessment of our article titled: "A Comparison Study of Deep Learning Methodologies For Music Emotion Recognition". Efforts were made to revise the manuscript to incorporate the suggestions provided by the reviewer's comments, which we answer individually below.

  1. Thanks for the comment. The response is broken down into points to make it clearer.
    1) Regarding the number of examples obtained for each class after augmentation. We applied each augmentation to all samples of a given dataset, assigning them the original classes. Hence, the number of classes is easily calculated (doubles in size). This happens for each of the mentioned transformations, so, for instance, for the Hybrid Augmented experiment, four audio transformations were considered, effectively obtaining four times more samples for each class (and the dataset). Information regarding the amount of synthesized samples was added to section 3.2.4 (lines 233-237).
    2) Regarding the augmentations used and the effect of these on the emotional content (the class) of the augmented sample, which might deviate from the original sample and associated class that is still being used. This is a very good point. Indeed, the identified problem may happen, just like it also happens in the more objective ML problems dealing with images, such as face recognition or optical character recognition. For instance, it is normal in those fields to rotate, mirror, and stretch the original images to make new training data. Extreme transformations will result in images that lose their original meaning, i.e., they no longer represent a valid human face or text character. The same applies here with music and emotion, where the augmented sample from an original happy sample might no longer carry the same happiness or even become noise/nonmusical. Here, as in other ML fields, this is not an issue because all these augmented samples will only be used to train the model, not being present in the test set, nor originating from samples in the test set. Thus, if the augmented samples become bad examples/noise, i.e., a sample that is no longer happy or an image that is no longer a human face or digit, the trained will be fed noise and learn worse, thus performing lower in the test set (original samples only). In summary, the augmentation data can indeed deviate from the associated labels, but such information is only used as a strategy during training, as discussed in the second comment's response, and as such, these effects will effectively be a penalty during the test. Hence, no issues with compromising the integrity of the results.
    3) No experiments or expert evaluation of the augmented data was conducted to assess its effect on the emotion classes. As mentioned in the previous point, this does not compromise in any way the integrity of our study and results since these are not used to test/validate any results. Still, this problem is indeed something that we may explore in the future since it will be a good contribution to the field - to understand the effect of each transformation and augmentation in MER. There are major hurdles to pursuing this goal. The idea of using expert evaluation is very time-consuming, and if human resources are available to annotate in the future, they will be best used to annotate original clips and extend the datasets. Thus, one possible strategy might be to design a future experiment with ML to assess that.
  2. There may have been some lack of clarity in our original text regarding the augmentation and data partitioning, thus wrongly inducing the reviewer. When splitting the train and test sets we only consider the original samples at first. After assigning samples to the corresponding sets, then we add the relevant synthesized samples to the train set only if the original sample is present there. This is not done for the test set were we only consider original samples.
    Thus, both sets are completely independent, with the train set containing original and synthesized samples, while the test set only contains original samples that were annotated and validated manually. Information regarding the train-test split procedure with augmented data was added to section 3.2.4 (lines 264-279).
  3. Thank you for your suggestions, however, the mentioned works were previously analyzed by the team and were not considered for the reasons discussed below. 
    The first mentioned work titled "MERGE Audio: Music Emotion Recognition next Generation-Audio Classification with Deep Learning" by Pedro Sá was conducted within our team. One of the methodologies proposed is reported to have reached 88% F1-score as pointed out by the reviewer. It is normal procedure in our group to replicate the previous work of team members, and, in the process, a bug in the implementation of the neural network was found, which inflated the performance of the model. After solving the bug and rerunning the experiment, the best possible results reached around 72% F1-score, falling short from the classical baseline of our work. Given the huge discrepancy of the reported results, the model and related experiments were not published as original intended.
    Furthermore, we have some concerns regarding the second work titled "A Music Playback Algorithm Based on Residual-Inception Blocks for Music Emotion Classification and Physiological Information". When discussing the training process in Section 4.4, it is not clear to us how the data split was conducted, as it is mentioned that a train-validate-test split was conducted, but later in Section 4.8, 10-fold cross-validation is mentioned, which in our view are not compatible. On one hand, should the data split have been conducted as mentioned in Section 4.4., this could lead to uneven distributions of emotion, genre, and other relevant data for Music Emotion Recognition. Without these considerations, the reported results may not be accurate if other datasets are used for further evaluation. On the other hand, 10-fold cross-validation is not enough to provide accurate metrics, since the same point discussed for train-validate-test split applies here. This is way we conduct our experiments with 10-repetitions 10-fold cross-validation, which ensures that a wide array of data distributions are tested.

Reviewer 2 Report

Comments and Suggestions for Authors

The authors have provided a comprehensive list of references, showcasing a strong foundation for their research. It would be beneficial to provide a brief summary or explanation of how each reference contributes to the study. [Page no. 16]

The funding sources for the research are clearly outlined, demonstrating transparency and accountability. It may be helpful to include a brief description of how the funding supported the research activities. [Page no. 15]

The authors have declared no conflicts of interest, which is important for maintaining the credibility and integrity of the research. It would be valuable to provide a statement on how potential conflicts of interest were managed throughout the study. [Page no. 15]

The disclaimer regarding responsibility for any potential consequences resulting from the content is a standard practice. It might be useful to expand on this disclaimer by including information on ethical considerations or guidelines followed during the research. [Page no. 16]

The segment-level approaches discussed in the article offer insights into enhancing the model's performance by breaking down samples into smaller segments. Providing more details on the results or implications of implementing these approaches would enrich the discussion. [Page no. 5]

Comments on the Quality of English Language

The English language used in the search results appears to be of high quality, with clear and concise writing throughout the document.

The authors demonstrate a strong command of English, as evidenced by the accurate grammar, punctuation, and sentence structure.

Technical terminology related to the field of Music Emotion Recognition (MER) is used appropriately and effectively, indicating a deep understanding of the subject matter.

The writing style is academic and professional, suitable for a research paper in the field of computer science and music technology.

Overall, the authors' proficiency in English contributes to the clarity and credibility of the research presented in the article.

Author Response

We sincerely thank the reviewer for the comments to our article titled: "A Comparison Study of Deep Learning Methodologies For Music Emotion Recognition". Efforts were made to revise the manuscript to incorporate the suggestions provided by the reviewer, which we answer individually below.

  1. We did our best to justify our references as they are cited throughout the paper, by giving some context or information and then giving the citation (which supports the sentence, hence their justification). For example, when introducing and discussing Gated Recurrent Units, the original paper proposing them is cited to give the original source of the approach and then justify our decision over using Long Short Term Memory units. In our perspective, adding a summary describing the purpose of each reference would increase considerably the length and hamper the readability of the article as a whole.
  2. The funding received did not impact the design or the conduction of our study. It was used to support the first and second authors' research grant as well as increase the amount and quality of available resources in the GPU server mentioned in Section 4.3. This was clarified in the Conflicts of Interest disclaimer in Page 16 (lines 505-506).
  3. The design of our study and the dataset used for the conducted experiments are all established by members of our team, with no conflicts of interest as stated in the Conflicts of Interest disclaimer in Page 15. Most of the software used was also developed by our team, with some adapted from publicly available repositories, which were credited throughout the study.
  4. We consider that there are no complex ethical issues to address. Regarding the annotation process for the New-4QAED dataset, we followed the usual approach of informing the annotators of the annotation purpose and the eventual public release of the dataset.
  5. It is not possible to directly compare the segment-level and sample-level approaches due to architectural differences, however, conclusions can be drawn regarding the advantages of leveraging smaller segments, such as learning more localized information in comparison to the more global information of sample-level models and the increased amount of training data resulting from splitting the larger samples. A mention to the point regarding learning features at the local-level was added to Section 3.2.2 (lines 178-179).

Reviewer 3 Report

Comments and Suggestions for Authors

This study conducted a comparative experiment encompassing various classical machine learning and deep learning approaches. The experimental results indicate that the integrated method, combining dense neural networks and convolutional neural network architectures, achieved the best performance with an F1 score of 80.20%, improving by approximately 5% compared to the best baseline result. This finding emphasizes the fusion of both handcrafted features and feature learning paradigms to fully leverage their advantages. However, there are several concerns and questions regarding the paper:

1.The paper evaluates different-sized datasets. Please provide a detailed analysis of how dataset size impacts performance.

2.Regarding the proposed Hybrid Augmented method, at what level is the information fusion performed? Could you provide more comparative experimental results to support its effectiveness?

3.In this study, are there any limitations in the application of techniques such as data augmentation and knowledge transfer?

4.Please explain the performance differences between the various methods presented in Tables 6 and 7, along with possible reasons for these differences.

5.Certain method selections and implementation details (e.g., the choice of GRU over LSTM, the determination of the fusion approach between CNN and DNN) require more explanation and justification.

Comments on the Quality of English Language

There are some spelling and grammatical errors in the paper (e.g., "conbining" should be "combining"). It is recommended to proofread the paper carefully.

Author Response

We sincerely thank the reviewer for the comments to our article titled: "A Comparison Study of Deep Learning Methodologies For Music Emotion Recognition". Efforts were made to revise the manuscript to incorporate the suggestions provided by the reviewer, which we answer individually below.

  1. The dataset size impact is discussed in some detail for each set of methodologies in Section 5. In Section 6, we discuss that we were able to verify that dataset size is more relevant than dataset size for most of the experiments conducted. For example, we can see in the Segment-level Architectures' results table (Table 4) that the New-4QAED complete set outperforms the balanced set by around 4% F1-score. The CRNN architecture's results in Table 3, despite not to the same extent, also present a significant difference of around 1.5% between the same sets. Future experiments should be conducted in larger datasets to further corroborate our findings. Further clarification regarding the impact of dataset size was added at the end of Section 6 (lines 475-479).
  2. Information fusion was done at the feature level, concatenating the processed handcrafted features, used as the input of the Dense Neural Network (DNN) portion, with the learned features from the Convolutional Neural Network (CNN). Beyond the comparative results reported using only the CNN architecture for classification, experiments utilizing only the DNN architecture were also conducted. These include not only the full 1714 feature set as in the Hybrid Augmented method, but also the top 100 features reported in the original study by Panda et al. However, these were not included as they did not reach the Support Vector Machine Baseline's performance, attaining 69.21% and 72.48%, respectively. The DNN-only experiments' descriptions were added to Section 3.2.1 (lines 162-165 and Figure 4), results were added to Table 3 and discussion was added to Section 5  (lines 391-396).
  3. To the best of our knowledge, there are no limitations beyond what is discussed in Sections 3.2.4 and 3.2.4. For data augmentation, we considered classical audio transformations and factors that would preserve the emotional content of the original samples as best as possible and that would not introduce noise into the training set. Still on data augmentation, the DeepSMOTE approach constrains the variation of the sample in a neighboring region in the same line as for the classic audio transformations. As for transfer learning, we considered tasks close to Music Emotion Recognition (MER) to maximize the relevance of each network's learned features.
  4. Regarding Table 6, the worst performing audio transformation, Background Noise and Tanh Distortion, can be explained by the answer to point 3, that these transformations considerably alter the emotional content of the original sample to the point that it is no longer the same. Still on Table 6, the poor performance of DeepSMOTE can be explained by too little variation of the generated samples when compared to the original sample, thus not adding any new information. Moreover, it can also be explained by changes made to the original Mel-spectrogram representation, which distorted important regions for classification.
    As for Table 7, the poor performance of all results can be due to multi-label classification learning many irrelevant features for MER, as explained in Section 5. Regarding the different datasets used to pre-train the CRNN, we can deduce that the datasets themselves present low quality, a common problem when using the Million Song Dataset. The low quality of this dataset is due to the user-generated labels used as annotations, which are not manually validated and are in many cases ambiguous, e.g., "love" can reference the underlying emotion of the song or that the user loves this song. Another possible reason is the data distribution regarding emotion, genre, and other relevant factors for MER. When building our datasets, we not only consider emotion when balancing each quadrant, but also genre, since some genres like metal and hip-hop are normally associated with angry emotions, which influences the model into learning that songs from one of these genres always fall into angry emotions. Especially when we consider that the three datasets considered were not originally built with MER in mind, this is a strong hypothesis for the under performance of these methodologies. These details were included in Section 5 when discussing each table's results (lines 405-407, 429-431 and 448-449).
  5. We thank the reviewer for pointing out the need for further justification. After further reviewing the article, we believe that many of the methods and implementation details are explained in some detail and that the original articles cited can complement these explanations. For the given examples, the choice of Gated Recurrent Units is justified in the paper that original proposed them, while the fusion approach in the Hybrid Augmented experiment was chosen to understand how handcrafted and learned features could complement themselves. More details regarding the examples provided can be found in Section 3.2.1 (lines 149-150 and 157-161).

Regarding Comments on the Quality of English Language: We thank the reviewer for pointing out these errors. To address this we have carefully proofread the current version of the manuscript.

Round 2

Reviewer 1 Report

Comments and Suggestions for Authors

All comments were taken into account by the authors, the edits made to the article are correct.